# Paired-Pulse Repetitive Trans-Spinal Magnetic Stimulation Supports Balance Ability While the Coil Orientation Significantly Determines the Effects: A Randomised, Placebo-Controlled Trial

**DOI:** 10.3390/biomedicines13081920

**Published:** 2025-08-06

**Authors:** Jitka Veldema, Michel Klemm, Jan Straub, Saskia Kurtzhals, Lea Sasse, Teni Steingräber

**Affiliations:** Faculty of Psychology and Sports Science, Bielefeld University, 33501 Bielefeld, Germany; michel.klemm@uni-bielefeld.de (M.K.); jan.straub@uni-bielefeld.de (J.S.); saskia.kurtzhals@uni-bielefeld.de (S.K.); lea.sasse@uni-bielefeld.de (L.S.); teni.steingraeber@uni-bielefeld.de (T.S.)

**Keywords:** paired-pulse repetitive magnetic stimulation, spinal cord, balance control, neural networks

## Abstract

**Objectives:** The primary objective was to investigate and compare the effects of three paired-pulse repetitive trans-spinal magnetic stimulation (PP-rTSMS) protocols on balance control and corticospinal network function. **Methods**: PP-rTSMS (800 pulses, frequency 100 Hz, intensity 70% of the resting motor threshold) was applied over the eighth thoracic vertebra (Th8) in twenty-seven young healthy individuals. Each proband received three verum sessions (using a verum coil with handle oriented (i) cranially, (ii) caudally, and (iii) laterally) and (iv) one sham session (using a sham coil) in a randomised order. Balance ability (Y Balance Test) and corticospinal network functions (motor evoked potentials (MEPs), cortical silent periods (SCPs)) were tested immediately (i) prior to and (ii) after each interventional session. **Results:** Each verum session induced a significant improvement in balance ability (cranially (F_1,26_ = 8.009; *p* = 0.009; η^2^ = 0.236), caudally (F_1,26_ = 4.846; *p* = 0.037; η^2^ = 0.157), and laterally (F_1,26_ = 23,804; *p* ≤ 0.001; η^2^ = 0.478) oriented grip) as compared to the sham session. In addition, the laterally oriented coil grip was associated with significantly greater balance benefits than both the cranial (F_1,26_ = 10.173; *p* = 0.004; η^2^ = 0.281) and caudal (F_1,26_ = 14.058; *p* ≤ 0.001; η^2^ = 0.351) grip orientations. No significant intervention-induced effects were detected on corticospinal network functions. **Conclusions:** Our data show that PP-rTSMS effectively supports balance control and that coil orientation significantly influences these effects. Further studies should test variations of this promising approach on healthy and disabled cohorts.

## 1. Introduction

Noninvasive magnetic stimulation is a popular therapeutic and research tool. Despite the enormous interest in this technique over the past two decades, its widespread transfer into clinical practice has not occurred. Inconclusive effects, interindividual variability, and the limited amount of evidence regarding a broad spectrum of application variations raise new questions about the effectiveness of this method. This study tested innovative stimulation protocols for posture and balance control.

### 1.1. Paired-Pulse Repetitive Magnetic Stimulation over the Spinal Cord

Existing applications of noninvasive neuromodulation in supporting balance control in both healthy and disabled cohorts primarily focus on the primary motor cortex (M1). In addition to the M1 region, the cerebellum, supplementary motor area, and dorsolateral prefrontal cortex have been targeted in several studies [1,2,3,4,5,6,7]. Other regions have rarely been considered [1,2,3,4,5,6,7]. Although several neuroimaging studies have demonstrated that a wide range of other cortical and subcortical brain regions and the brainstem are crucially involved in balance control [8,9], existing technical solutions are insufficient to effectively target deep subcortical and brainstem structures. Therefore, the spinal cord is a promising alternative to the brain. Extensive data indicates that the spinal cord is crucially involved in balance tasks [10,11,12]. For example, the relationship between improved balance and suppressed Hoffmann reflex has been repeatedly described in both healthy and disabled cohorts [10,11,12]. A theory originating in the early twentieth century proposes that so called “central patterns generators” (CPGs) are core elements of posture and balance control [13,14,15]. These self-regulating neural circuits (located within the lumbar and cervical spinal cord intumescences) function under the control of the brainstem without input from higher brain regions. It is assumed that CPGs regulate optimal agonist–antagonist interactions during stereotypical rhythmic motor patterns (such as walking, cycling, or swimming) and dampen the disturbing effects of overshooting reflexive reactions (e.g., during stumbling) by facilitating or inhibiting alpha motoneuron activity [13,14,15]. Based on these assumptions [10,11,12,13,14,15], the direct modulation of the spinal cord by either direct current stimulation (DCS), alternating current stimulation (ACS), or repetitive magnetic stimulation (rMS) may generate balance and postural control benefits. This expectation is supported by our previous studies, which demonstrated that both anodal DCS and ACS over the spinal cord support balance ability in healthy young people [16,17], and their effects are not weaker than the effects of cranial applications [16]. In the present study, we investigated the effects of spinal rMS in a young healthy cohort.

Pulse frequency and pattern are considered key determinants of stimulation-evoked effects. The popularity of non-patterned high- and low-frequency protocols, as well as patterned continuous and intermittent theta burst stimulation protocols (cTBS, iTBS), is evident not only in the field of balance ability [1,2,3,4,5,6,7] but also in other types of neurobehavioural and neurocognitive research [18,19,20,21]. Many other protocols, such as paired-pulse repetitive magnetic stimulation (PP-rMS) [22,23,24,25,26] or quadripulse repetitive magnetic stimulation (QPS-rMS) [27,28], have not been extensively investigated. Our study will foster an understanding of PP-rMS protocols.

### 1.2. The Influence of Coil Orientation on Stimulation-Induced Effects

The orientation of the stimulation coil is a factor that may significantly determine the effects of rMS on neural networks and abilities. The majority of studies in this field tested the effects of different coil orientations on M1 [29,30,31,32,33,34]. Their data suggested that a coil placed perpendicular to the central sulcus (at an angle of 45° with respect to the sagittal plane) modulates neural networks more effectively than other coil positions [29,30,31,32,33,34]. These findings significantly impact existing research and practice because a coil oriented perpendicular to the central sulcus is used in almost all rMS applications over the M1 or other frontal lobe regions [35,36]. Although, it is questionable whether the optimal coil position as detected for the M1 region is transferable to other brain or spinal cord regions. For example, the neural architecture (orientation of neurons in regard to skull surface) or the physical properties of the overlying bones (e.g., thickness) significantly differ across different regions [37,38]. This may impact the distribution of electrical current through the neural issue. Systematic research on this field may contribute to the development of more efficient stimulation protocols for several cohorts. To date, only a few studies have examined the effects of coil orientation outside the M1 area. Their results showed that rMS over the visual [39,40,41] or prefrontal [42,43] cortex regions may induce coil orientation-dependent effects on neural networks [39,40,41,42,43], as well as on behavioural [39] and cognitive [43] abilities. Studies testing the effects of spinal rMS are rare, and to date, no direct comparison of diverse coil orientations for spinal applications exists. However, the available data indicate that spinal MS is safe, well tolerated, and an effective method for modulating neural networks in both healthy and disabled cohorts [44], also indicating that it improves gait in Parkinson’s disease [45] and spasticity in multiple sclerosis [46]. Our study fills the gap in the evidence in this field and compares the effects of three different coil orientations on the thoracal spinal cord. Even if positive effects could be expected independently of coil positioning, the effect sizes can significantly differ across interventions.

## 2. Methods

### 2.1. Study Design

This crossover study investigated and compared the effects of three different paired-pulse repetitive trans-spinal magnetic stimulation (PP-rTSMS) protocols on balance control and neural network function in healthy young people. Three verum interventional sessions with the stimulation coil grip oriented (i) cranially, (ii) caudally, or (iii) laterally (rightward), as well as (iv) one placebo session, were applied to each subject in a randomised (machine-generated) order. The washout period between sessions was at least five days. Balance ability and corticospinal network function were tested immediately before and after each intervention. Adverse effects were recorded after each intervention. Both the participants and the examiner were blinded to interventio n allocation. This study was performed in line with the standards established by the Declaration of Helsinki, approved by the Ethics Committee of Bielefeld University (EUB-2024-060), and registered in the German Clinical Trial Register on 26 May 2025 (DRKS00037004). The manuscript was written in accordance with the recent CONSORT guidelines [47].

### 2.2. Participants

Individuals who met the following criteria were included: (1) age between 18 and 30 years; (2) no contraindications for noninvasive rMS [48]; and (3) no relevant neurological, psychiatric, or orthopaedic disorders. All participants provided written informed consent before participation. Sample size calculation using G*power (version 3.1.9.7) analysis (effect size = 0.35, α error probability *p* < 0.05, Power = 0.95) indicated that a minimum of 27 subjects is required to detect statistically significant effects using ANOVA with four interventions and two time points.

### 2.3. Assessments

#### 2.3.1. Balance Control

The Y Balance Test (YBT) [49] was applied using a test kit (FMS, Chatham, MA, USA). The maximum reach distance of the free lower leg in the (a) anterior, (b) posterolateral, and (c) posteromedial directions was measured during a one-leg stance on the opposite leg. Five trials were conducted for each leg in each direction. The mean values were used for the analysis. A greater reach distance indicates better balance ability.

#### 2.3.2. Neurophysiological Measures

A biphasic magnetic stimulator (PowerMAG QPS) was used with a double-cone coil placed parallel to the sagittal plane (MAG & More GmbH, Munich, Germany) and silver–silver chloride electrodes positioned in a belly tendon technique. The participants sat on the transcranial magnetic stimulation (TMS) chair. The motor hotspots and resting motor thresholds of both tibialis anterior (TA) muscles were determined using standardised procedures [50,51]. MEPs and CSPs were tested from both TA muscle hotspots with constant stimulator output throughout all sessions. MEPs were recorded from relaxed TA muscles during 30 single TMS pulses (intensity 110% of the resting motor threshold (rMT)) over the contralateral TA hotspot. CSPs were recorded from activated (20–30% of maximal voluntary contraction) TA muscles during 15 single TMS pulses (intensity 130% of rMT) over the contralateral TA hotspot. MEP and CSP amplitudes and durations were measured [50,51], and mean values were calculated.

#### 2.3.3. Adverse Effects

A tolerability questionnaire was used to evaluate the occurrence of adverse effects (such as fatigue, headache, neck pain, light-headedness, and nausea) and rate their severity (from mild (1) to severe (5)) [52].

### 2.4. Intervention

A biphasic magnetic stimulator (PowerMAG QPS), with an actively cooled figure-eight coil and a sham figure-eight coil (MAG & More GmbH, Munich, Germany) was used. The participants lay face-down on a therapy table. PP-rTSMS (frequency 100 Hz, intensity 70% rMT, 400 PP with a 2 s interval in between) was applied over the Th8. Palpation was used to identify the targeted region [53,54]. Verum interventions were applied through the actively cooled figure-eight coil positioned horizontally, with the coil handle pointing (a) cranially, (b) caudally, or (c) laterally rightward. Sham stimulation was applied with a sham figure-eight coil oriented in one of the three directions (Figure 1) chosen randomly. The sham coil induces no or only a negligible electric field in the brain but produces similar sound and vibration effects to a real coil, and neither probands or an investigator can distinguish between them [55].

### 2.5. Analysis

SPSS software package version 27 (International Business Machines Corporation Systems, Chicago, IL, USA) was used to analyse the data collected during the study. Repeated measure ANOVAs with the factors “intervention” and “time” were used to compare the pre–post changes across interventions. The partial eta squared method was used to measure the effect size (η^2^ ≥ 0.01 “small”, η^2^ ≥ 0.06 “medium”, η^2^ ≥ 0.14 “large”) [56]. The Kolmogorov–Smirnov normality test, as well as Mauchly’s sphericity tests and Bonferroni corrections, was applied. Independent sample *t*-tests were used to check the pre-interventional comparability across interventions. *p*-values of ≤ 0.05 were considered statistically significant.

## 3. Results

In total, 28 students were enrolled. One proband was lost at follow-up (without giving a reason), and its data were removed from the analysis. The remaining 27 subjects (age 25.2 ± 3.4 years, 13 females, 14 males, 22 right-footed, 5 left-footed) completed all interventions. The foot that is favoured when kicking a ball was considered dominant [57]. Table 1 presents the balance and corticospinal measurement data collected during the experiments. The pre-interventional evaluations show no significant differences.

### 3.1. Y Balance Test

Figure 2 presents the intervention-induced changes. The total balance ability score improved significantly with real PP-rTSMS applied with all coil orientations (cranially (F_1,26_ = 8.009; *p* = 0.009; η^2^ = 0.236), caudally (F_1,26_ = 4.846; *p* = 0.037; η^2^ = 0.157), and laterally (F_1,26_ = 23,804; *p* ≤ 0.001; η^2^ = 0.478)) compared to sham stimulation. In addition, the lateral coil grip orientation was associated with greater benefits than both the cranial (F_1,26_ = 10.173; *p* = 0.004; η^2^ = 0.281) and caudal (F_1,26_ = 14.058; *p* ≤ 0.001; η^2^ = 0.351) coil grip orientations.

The balance ability of the right leg improved significantly with real PP-rTSMS applied with the coil handle oriented cranially (F_1,26_ = 4.821; *p* = 0.037; η^2^ = 0.156) and laterally (F_1,26_ = 22.191; *p* ≤ 0.001; η^2^ = 0.460) compared with sham stimulation. In addition, the lateral coil grip orientation was associated with a stronger improvement than the cranial (F_1,26_ = 7.344; *p* = 0.012; η^2^ = 0.565) and caudal (F_1,26_ = 15.034; *p* ≤ 0.001; η^2^ = 0.366) coil grip orientations.

The balance ability of the left leg improved significantly with real PP-rTSMS applied through a coil with a cranially (F_1,26_ = 9.486; *p* = 0.005; η^2^ = 0.267), caudally (F_1,26_ = 7.382; *p* = 0.012; η^2^ = 0.221), and laterally (F_1,26_ = 23.314; *p* ≤ 0.001; η^2^ = 0.473) oriented grip compared to sham stimulation. In addition, the lateral coil grip orientation was associated with greater improvement than the cranial (F_1,26_ = 12,424; *p* = 0.002; η^2^ = 0.323) and caudal (F_1,26_ = 8.966; *p* = 0.006; η^2^ = 0.256) coil grip orientations.

### 3.2. Neurophysiological Measures

No significant intervention-induced effects were observed in MEP and CSP amplitudes and durations.

### 3.3. Adverse Effects

No severe adverse events were reported. The occurrence of fewer severe adverse events is shown in Table 2. No significant differences were detected between the conditions.

## 4. Discussion

Our study showed that PP-rTSMS is a safe, well-tolerated, and effective method for supporting balance ability in healthy young people. The stimulation coil orientation plays an important role in the stimulation-induced effects. Real PP-rTSMS was associated with significant behavioural improvement compared to sham, regardless of the coil orientation. A laterally oriented coil grip was associated with significantly greater benefits than a cranially or caudally oriented coil grip.

### 4.1. PP-rTSMS Is a Promising Approach to Support Human Abilities

Despite the central role of the brain, the spinal cord is increasingly becoming the focus of noninvasive neuromodulation research. Several cohorts may benefit from this approach in the future. It has been repeatedly demonstrated that high-frequency rTSMS can induce neural, postural, and/or locomotor benefits in disabled cohorts [58,59,60,61]. For example, the application of both 10 Hz rTSMS [58] and 20 Hz rTSMS [59] over two weeks supported the recovery of neural tissue [58,59], together with locomotor recovery [58], after spinal cord injury. Ten sessions of 20 Hz rMS over C7 improved postural control and functional ambulation in patients with relapsing–remitting multiple sclerosis [60]. Eight sessions of 5 Hz rMS over Th12-L1 supported motor rehabilitation in Parkinson’s disease [61]. Despite these encouraging results, the existing evidence is insufficient, and numerous additional studies are needed before noninvasive spinal modulation can be established in the framework of evidence-based medicine. Our study enhances the evidence in this field and shows that PP-rTSMS applied over L2 is safe, well-tolerated, and effective in supporting balance control in healthy people. Future studies should test this protocol in patient cohorts.

The reason PP-rMS protocols have rarely been investigated is unknown [22,23,24,25,26]. It is possible that a temporary unpleasant/painful sensory experience during cranial PP-rMS (PP-rTMS) is responsible for the infrequent use of this approach. Similar discomfort was not observed for spinal PP-rMS in the present study, and this is a crucial advantage of using spinal rMS. Previous experiments have demonstrated that cranial PP-rMS successfully modulates neural networks [22,23,24,25,26] and visuomotor skills [22,25] in healthy people [22,23,24,25], as well as in patients with Parkinson’s disease [26], and induces no relevant adverse effects [22,23,24,25,26]. PP-rTMS delivered at 667 Hz [22,23,24,25,26], 250 Hz [24], 222 Hz [22,23,24], and 200 Hz [24] every 5 s increased corticospinal excitability [22,24,26] and improved hand motor function [22] in healthy subjects. The application of 100 Hz and 333 Hz PP-rTMS applied every second increased (100 Hz) and decreased (333 Hz) corticospinal excitability in patients with Parkinson’s disease and healthy individuals [25] and improved hand function in Parkinson’s patients [25]. PP-rTMS delivered every 1.7 s at 100 Hz decreased corticospinal excitability and increased short interval intracortical inhibition and CSPs in healthy people [23]. Our study enhances the evidence in the field of PP-rMS protocols and shows that 100 Hz PP-rMS applied every two seconds has positive behavioural effects. The MEP and CSP evaluations did not detect any effects on the corticospinal networks, which is inconsistent with previous observations [22,23,24,25,26]. Whether other neuroimaging techniques can be more successful in this regard remains an open question.

### 4.2. Coil Orientation Significantly Influences Stimulation-Induced Effects

Several theories exist regarding the impact of stimulation coil orientation on stimulation-induced effects [31,35]. One theory suggests that the orientation of neurons in relation to the current direction plays a crucial role [37,62,63]. It is assumed that current applied radially inwards along the somatodendritic axis (from the dendrite to the soma) causes depolarisation, current applied radially outwards (from the soma to the dendrite) causes hyperpolarisation, and current applied tangentially across the somatodendritic axis causes little to no polarisation effect [37,63].

It is assumed that the current flow induced by MS within neural networks is in the opposite direction of the initial current flow within the coil windings [35]. In the framework of this physical phenomenon and the construction of our equipment, the current produced at the centre of the figure-of-eight coil flows towards the coil grip. Thus, stimulation applied with the grip oriented cranially should increase the neural activity of the ascending (sensory) spinal cord pathways above Th8 and decrease the neural activity of the descending (motor) spinal cord pathways above Th8. The caudal grip orientation should be associated with the opposite effects (decreased neural activity of sensory and increased neural activity of motor spinal cord pathways) below Th8. Stimulation applied with a grip coil oriented laterally may more effectively modulate sensory and motor neurons in the periphery that enter and exit the spinal cord via the dorsal and ventral horns and run in parallel to the induced current.

Another theory emphasises that the type of tissue crucially determines the distribution of an electric field throughout the human body. Generally, tissues with higher water proportions (neural tissue, muscles, and blood [64]) have better conductivity than those with lower water proportions (bones and adipose tissue [64]). This may explain the inferior effectiveness of stimulation through a coil with a cranially and caudally oriented grip, which induces current flow towards the vertebral arches with bony processes. In contrast, a stimulation coil with a laterally oriented grip directly targets the muscles and nerves lying lateral to the vertebral bodies.

## 5. Conclusions

The present study showed that PP-rTSMS is a safe, well-tolerated, and highly effective method for balance control. The generalisability of our findings is limited by the relatively small sample size. Future studies should investigate variations in this promising protocol in larger cohorts with different ages and health conditions. Patients with stroke, multiple sclerosis, spinal cord injury, Parkinson’s disease, and other illnesses may benefit from this approach. Application of additional follow-up evaluations may contribute to a better understanding of short- and long-term behavioural and neural effects.

## Figures and Tables

**Figure 1 biomedicines-13-01920-f001:**
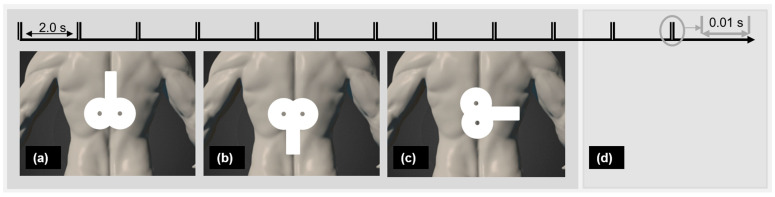
Real PP-rTSMS with coil grip oriented (**a**) cranially, (**b**) caudally, and (**c**) laterally and (**d**) placebo PP-rTSMS were applied to each proband in a randomised order. Notes: PP-rTSMS = paired-pulse repetitive trans-spinal magnetic stimulation; s = second.

**Figure 2 biomedicines-13-01920-f002:**
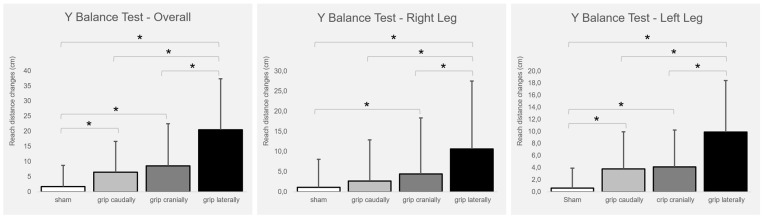
Intervention-induced changes (means and SD) of balance ability (overall, right leg, left leg) in relation to the baseline. Notes: cm = centimetre; * *p* ≤ 0.05.

**Table 1 biomedicines-13-01920-t001:** Balance ability and corticospinal networks functions (means and SD) at both time points (pre and post).

					Real PP-rTSMS Coil Grip Cranially	Real PP-rTSMS Coil Grip Caudally	Real PP-rTSMS Coil Grip Laterally	Sham PP-rTSMS
YBT (cm)		total		pre	504 ± 51	503 ± 50	503 ± 47	496 ± 45
				post	513 ± 50 ^*S^	510 ± 48 ^*S^	523 ± 50 ^*S,Ca,Cr^	497 ± 44
		right leg	pre		251 ± 28	251 ± 26	251 ± 24	247 ± 24
			post		256 ± 27 ^*S^	254 ± 25	262 ± 26 ^*S,Ca,Cr^	248 ± 24
		left leg	pre		253 ± 24	252 ± 24	252 ± 23	249 ± 22
			post		257 ± 23 ^*S^	256 ± 24 ^*S^	261 ± 25 ^*S,Ca,Cr^	249 ± 22
MEP	Amplitude (mV)	right leg	pre		0.70 ± 0.57	0.61 ± 0.45	0.64 ± 0.66	0.71 ± 0.94
			post		0.68 ± 0.51	0.60 ± 0.45	0.63 ± 0.58	0.70 ± 0.59
		left leg	pre		711 ± 600	540 ± 466	740 ± 745	492 ± 294
			post		678 ± 492	558 ± 428	787 ± 754	534 ± 416
	Duration (ms)	right leg	pre		23.0 ± 4.0	21.9 ± 2.5	21.8 ± 4.7	22.4 ± 2.9
			post		22.8 ± 5.0	21.8 ± 2.7	22.1 ± 4.2	22.9 ± 2.7
		left leg	pre		22.7 ± 2.9	22.4 ± 3.1	22.2 ± 3.1	21.9 ± 3.6
			post		22.5 ± 3.5	22.2 ± 3.2	23.1 ± 3.2	21.7 ± 4.6
CSP	Amplitude (mV)	right leg	pre		3.94 ± 1.39	3.80 ± 1.43	3.69 ± 1.54	3.76 ± 1.52
			post		4.00 ± 1.50	3.59 ± 1.43	3.65 ± 1.51	3.53 ± 1.34
		left leg	pre		3.57 ± 1.41	3.13 ± 1.30	3.62 ± 1.65	3.29 ± 1.52
			post		3.22 ± 1.21	3.09 ± 1241	3.51 ± 1.41	3.03 ± 1.25
	Duration (ms)	right leg	pre		186 ± 50	185 ± 46	174 ± 40	181 ± 45
			post		192 ± 44	191 ± 41	189 ± 46	187 ± 40
		left leg	pre		180 ± 42	186 ± 47	177 ± 41	189 ± 53
			post		190 ± 39	198 ± 30	193 ± 43	198 ± 49

Notes: cm = centimetre; CSP = cortical silent period; MEP = motor evoked potential; ms = millisecond; mV = millivolt; PP-rTSMS = paired-pulse repetitive trans-spinal magnetic stimulation; YBT = Y Balance Test; ^Ca^ = significant intervention-induced changes in comparison to real PP-rTSMS with coil grip oriented caudally; ^Cr^ = significant intervention-induced changes in comparison to real PP-rTSMS with coil grip oriented cranially; ^S^ = significant intervention-induced changes compared to sham PP-rTSMS; ^*^ = *p* ≤ 0.05.

**Table 2 biomedicines-13-01920-t002:** Occurrence (number of probands) and severity (mean and SD; total amount) of adverse effects after each intervention.

	Real PP-rTSMS Coil Grip Cranially			Real PP-rTSMS Coil Grip Caudally			Real PP-rTSMS Coil Grip Laterally			Sham PP-rTSMS		
	Nr.	Severity		Nr.	Severity		Nr.	Severity		Nr.	Severity	
		Mean	Total		Mean	Total		Mean	Total		Mean	Total
Fatigue	11	1.3 ± 0.7	14.0	10	2.1 ± 1.1	21.0	10	2.2 ± 1.0	22.0	10	1.8 ± 1.1	18.0
Headache	9	1.3 ± 1.0	12.0	5	1.6 ± 0.9	8.0	5	1.6 ± 0.5	8.0	6	1.5 ± 0.8	9.0
Neck pain	4	1.5 ± 0.6	6.0	3	1.7 ± 0.6	5.0	2	2.0 ± 0.0	4.0	2	2.0 ± 1.4	4.0
Lightheadedness	3	1.33 ± 0.6	4.0	3	1.3 ± 0.6	4.0	4	1.5 ± 0.6	6.0	2	2.0 ± 1.4	4.0
Unplesant tingling	2	1.0 ± 0.0	2.0	1	2.0 ± 0.0	2.0	2	1.8 ± 0.4	3.5	1	1.0 ± 0.0	1.0
Toothache	2	1.0 ± 0.0	2.0	1	1.0 ± 1.0	1.0	-	-	-	3	1.7 ± 1.2	5.0
Nausea	1	1.0 ± 0.0	1.0	1	1.0 ± 0.0	1.0	-	-	-	1	1.0 ± 0.0	1.0
Burning sensation	1	2.0 ± 0.0	2.0	-	-	-	-	-	-	-	-	-
Itching	1	3.0 ± 0.0	3.0	-	-	-	-	-	-	-	-	-
Ringing in ears	-	-	-	-	-	-	1	4.0 ± 0.0	-	-	-	-
Hearing problems	-	-	-	-	-	-	-	-	-	-	-	-
Vision problems	-	-	-	-	-	-	-	-	-	-	-	-
Others problems	-	-	-	-	-	-	-	-	-	-	-	-
	34	1.4 ± 0.14	46.0	24	1.8 ± 0.30	42.0	23	2.1 ± 0.33	47.5	25	1.7 ± 0.24	42.0

Notes: Nr. = number of probands; PP-rTSMS = paired-pulse repetitive trans-spinal magnetic stimulation.

## Data Availability

The datasets generated and/or analysed in the current study are available from the corresponding author upon reasonable request.

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
