# Peer review of "Paired-Pulse Repetitive Trans-Spinal Magnetic Stimulation Supports Balance Ability While the Coil Orientation Significantly Determines the Effects: A Randomised, Placebo-Controlled Trial"

_biomedicines, 2025, doi:10.3390/biomedicines13081920_

Round 1
Reviewer 1 Report (Previous Reviewer 2)
Comments and Suggestions for Authors
very interesting research but it has been improved:
add a flow chart in consort style fo your study
tablas are images, please substitute by editable versions
perform normality test (shapiro test) to evaluate distribution and then choose bette the parametric anovia you used or a non-parametric version, in fact, you have pre-post with 3 groups so youa have choos bettween a two way anova or Kruskall-wallis test, and posthoc pairwise comparison must been calculated with boferroni corrrection
add table 1 with sample baseline characteristicas and p value to demonstrate the absence of baseline differences
outcome table have been reorganised:
lin1 1 variables names, line 2 and 3 pre and post in group 1, lines 4, 5 6 and 7 the same for the other 2 groups, line 8 p valules and line 9 effect size with 95%CI
add post hoc tables ofr significant omnibus variables differences
Author Response
Comment 1: Add a flow chart in consort style for your study
Answer: Thank you for your comment. We acknowledge that a flow chart diagram may improve the clarity over the course of the study. However, it is difficult to make an appropriate diagram for a crossover study with four interventions. We included information to this topic within the main text.
Results chapter: In total, 28 students were enrolled. One proband was lost at follow up (without giving reason) and its data were removed from the analysis. The remaining 27 subjects (age 25.2 ± 3.4 years, 13 females, 14 males, 22 right-footed, 5 left-footed) completed all interventions.
Comment 2: Tables are images, please substitute by editable versions.
Answer: Thank you for this hint. We submitted the Tables in Excel.
Comment 3: perform normality test (shapiro test) to evaluate distribution and then choose bette the parametric anovia you used or a non-parametric version, in fact, you have pre-post with 3 groups so youa have choos bettween a two way anova or Kruskall-wallis test, and posthoc pairwise comparison must been calculated with boferroni corrrection
Answer: We used the repeated measures ANOVA with the factors “intervention” and “time” to compare the pre-post changes across interventions. The SPSS permits only Greenhouse-Geisser corrections (but not Bonferroni correction) by using of this test. Similarly, the Kolmogorov-Smirnov normality test and Mauchly's sphericity tests are standard settings by using of this (repeated measures ANOVA with the factors “intervention” and “time”) tool.
Analysis chapter: SPSS software package version 27 (International Business Machines Corporation Systems) was used to analyse the data collected during the study. Repeated measure ANOVAs with the factors “intervention” and “time” were used to compare the pre-post changes across interventions. The Kolmogorov-Smirnov normality test, Mauchly's sphericity tests and Greenhouse-Geisser corrections were applied. Independent sample t-tests were used to check the pre-interventional comparability across interventions.
Comment 4: add table 1 with sample baseline characteristicas and p value to demonstrate the absence of baseline differences
Answer: The baseline values (balance, electrophysiology) for each intervention are contained in the Table 1. The statistical analysis did not detect differences between interventions for pre-interventional evaluations. This information was included within the main text. The sample description (age, sex etc.) did not differ across interventions, due to the crossover study design. Its description is included within the main text.
Results chapter: The pre-interventional evaluations show no significant differences.
Results chapter: 27 subjects (age 25.2 ± 3.4 years, 13 females, 14 males, 22 right-footed, 5 left-footed)
Comment 5: outcome table hav been reorganised: lin1 1 variables names, line 2 and 3 pre and post in group 1, lines 4, 5 6 and 7 the same for the other 2 groups, line 8 p valules and line 9 effect size with 95%CI. add post hoc tables ofr significant omnibus variables differences
Answer: Dear reviewer, thank you very much for your suggestion. However, we would maintain the outline of the Table. The comparison of four interventions (cranially, caudally, laterally, sham), three assessments (YBT, MEP,CSP) and two legs (right, left, total) would leads to 120 additional values (40 p values, 40 effect size values and 80 CI values). This is too much and the table would lose its clarity. The statistics of significant intervention-induced effects are included within the main text.
Results chapter: The total balance ability score improved significantly with real PP-rTSMS applied with all coil orientations (cranially (F1,26 = 8.009; p = 0.009; η² = 0.236), caudally (F1,26 = 4.846; p = 0.037; η² = 0.157), and laterally (F1,26 = 23,804; p ≤ 0.001; η² = 0.478) oriented grip) compared to sham stimulation. In addition, lateral coil grip orientation was associated with greater benefits than both cranial (F1,26 = 10.173; p = 0.004; η² = 0.281) and caudal (F1,26 = 14.058; p ≤ 0.001; η² = 0.351) coil grip orientations.
The balance ability of the right leg improved significantly with real PP-rTSMS applied with the coil handle oriented cranially (F1,26 = 4.821; p = 0.037; η² = 0.156) and laterally (F1,26 = 22.191; p ≤ 0.001; η² = 0.460) compared with sham stimulation. In addition, the lateral coil grip orientation was associated with a stronger improvement than the cranial (F1,26 = 7.344; p = 0.012; η² = 0.565) and caudal (F1,26 = 15.034; p ≤ 0.001; η² = 0.366) coil grip orientations.
The balance ability of the left leg improved significantly with real PP-rTSMS applied through a coil with a cranially (F1,26 = 9.486; p = 0.005; η² = 0.267), caudally (F1,26 = 7.382; p = 0.012; η² = 0.221), and laterally (F1,26 = 23.314; p ≤ 0.001; η² = 0.473) oriented grip compared to sham stimulation. In addition, the lateral coil grip orientation was associated with greater improvement than the cranial (F1,26 = 12,424; p = 0.002; η² = 0.323) and caudal (F1,26 = 8.966; p = 0.006; η² = 0.256) coil grip orientations.

Reviewer 2 Report (Previous Reviewer 1)
Comments and Suggestions for Authors
Thank you for improvement of quality of article. However, some problems are still observed in the manuscript.
It seems that with the effect size of 35% and power of 95% and ANOVA analysis, the sample size should be more than 27. Approximately more than 50. Why do you determine 27 sample in each group?
Please explain with details about the blinfding of assessor and participants in the method section.
Author Response
Comment: It seems that with the effect size of 35% and power of 95% and ANOVA analysis, the sample size should be more than 27. Approximately more than 50. Why do you determine 27 sample in each group?
Answer: Dear reviewer, the sample size calculation using G*power (version 3.1.9.7) (effect size = 0.35, α error probability p < 0.05, Power = 0.95) revealed that 27 subjects are sufficient to detect statistically significant effects using ANOVA with four interventions and two time points.
Comment: Please explain with details about the blinding of assessor and participants in the method section.
Answer: Thank you for this suggestion. We included the information within the main text.
Methods chapter: Verum interventions were applied through the actively cooled figure-8 coil positioned horizontally, with the coil handle pointing (a) cranially, (b) caudally, or (c) laterally rightward. Sham stimulation was applied with a sham figure-8 coil oriented in one of the three directions (Figure 1) chosen randomly. The sham coil induces no or only a negligible electric field in the brain, but produces similar sound and vibration effects as real coil and neither probands or investigator can distinguish between then [55].

Round 2
Reviewer 1 Report (Previous Reviewer 2)
Comments and Suggestions for Authors
your comment are right, respect to bonferroni correction, this must be applied to posthoc pairwise comparison, t-test you applied afetr omnibus test to detect between which specficif grups are differences
Author Response
Comment: your comment are right, respect to bonferroni correction, this must be applied to posthoc pairwise comparison, t-test you applied afetr omnibus test to detect between which specficif grups are differences
Answer: We revised the manuscript in line with this comment
Analysis chapter: … and Bonferroni corrections were applied.
This manuscript is a resubmission of an earlier submission. The following is a list of the peer review reports and author responses from that submission.
Round 1
Reviewer 1 Report
Comments and Suggestions for Authors
Please consider the bellow comments:
Abstract:
-Please add the P value and effect size in the results section.
Introduction:
-Please describe with more details about the necessary of the comparing effects rTMS coil’s positioning on the considered variables.
-please add the hypotheses of your study based on the aim.
Methodology
-Please declare the method of group randomization.
-Please add the code of clinical trial registration in this section.
-Please explain about the blinding level of this study. Do you consider the blinding of participants, assessor or therapist? More information on the blinding would enhance the validity of the study design.
-One of the main limitations of this study is that the participants were healthy and without any postural impairments. Then, the results of this study cannot generalize to patients with real postural disorders. It is better to enter the patients with postural disorder to indicate the clinical value of the intervention.
-The sample size is relatively small, which may limit the generalizability of the findings. You should describe with details how this sample size is determined.
- Do the authors consider the level of physical activity of the participants in the inclusion criteria? This item is important factor in the neural network.
-Y balance test as a subjective test has not high accuracy for detection the balance and postural ability. The authors could use some postural tests such as the postural stability indices on Byodex system or force plate to assess the posture objectively.
Results
-The results are clearly presented, with appropriate use of statistical analysis to support the findings. Please add the mean age and rate of sex in the table 1 for each group and also report the between group analysis of these variables.
- it is better to consider a follow up assessing to investigate the long-term effects of interventions.
-Please add the effect size of balance differences among group in the table 1.
Discussion
The discussion needs to be interpreted the results and relates them to existing literature more details.
Why the findings just indicated improvement of balance ability after intervention, without any changes in MEP and CSP. The reason of this findings should be explaining in the discussion. It seems that the interventions were not changes in neural network among groups. It may be to not shown any differences among groups even in the balance test, if this assessed long-termly.
Please add the suggestions and limitations for future research. The potential impact of limitations of this study on the study's conclusions should be more explicitly discussed.
Author Response
Dear reviewer, thank you for the taking the time to read our manuscript and to make improvement suggestions. We revised the manuscript carefully in line with them. Our answers and modifications can be find below. We hope very much that our manuscript is now suitable for publication in Biomedicines.
Comment 1:_____________________________________________________________________
Abstract:
-Please add the P value and effect size in the results section.
Response 1: Thank you for this input. We revised the abstract in line with them
Line 37: Each verum session induced a significant improvement in balance ability (cranially (F1,26 = 8.009; p = 0.009; η² = 0.236), caudally (F1,26 = 4.846; p = 0.037; η² = 0.157), and laterally (F1,26 = 23,804; p ≤ 0.001; η² = 0.478) oriented grip) as compared to the sham session. In addition, the laterally oriented coil grip was associated with significantly greater balance benefits than both the cranial (F1,26 = 10.173; p = 0.004; η² = 0.281) and caudal (F1,26 = 14.058; p ≤ 0.001; η² = 0.351) grip orientations.
Comment 2:_____________________________________________________________________
Introduction:
-Please describe with more details about the necessary of the comparing effects rTMS coil’s positioning on the considered variables.
Response 2: Thank you for this hint. We added this information. Beside this, this topic is comprehensively discussed in the Discussion chapter.
Line 99: Although, it is questionable, if the optimal coil position as detected for the M1 region is transferable to other brain or spinal cord regions. E.g., the neural architecture (orientation of neurons in regard to skull surface) or the physical properties of the overlying bones (e.g., thickness) significantly differ across different regions [37,38]. This may impact the distribution of electrical current through the neural issue. Systematic research on this field may contribute to develop more efficient stimulation protocols for several cohorts.
Comment 3:_____________________________________________________________________
-please add the hypotheses of your study based on the aim.
Response 3: We added a sentence to the content required.
Line 114: Even if positive effects could be expected independently on coil positioning, the effect sizes can significantly differ across intervention.
Comment 4:____________________________________________________________________
Methodology
-Please declare the method of group randomization.
Response 4: Computer-based randomisation was used to allocate the interventions in a randomized order. This information is included in the Study design chapter
Line 123: randomised (machine-generated) order
Comment 5:____________________________________________________________________
-Please add the code of clinical trial registration in this section.
Response 5: Thank you for this hint. We added this information in Study design chapter.
Line 129: ... registered in the German Clinical Trial Register on 26 May 2025 (DRKS00037004).
Comment 6:____________________________________________________________________
-Please explain about the blinding level of this study. Do you consider the blinding of participants, assessor or therapist? More information on the blinding would enhance the validity of the study design.
Response 6: Thank you for this relevant hint. We added this information in the Study design chapter.
Line 126: Both, the participants and the examiner were blinded to intervention allocation.
Comment 7:____________________________________________________________________
-One of the main limitations of this study is that the participants were healthy and without any postural impairments. Then, the results of this study cannot generalize to patients with real postural disorders. It is better to enter the patients with postural disorder to indicate the clinical value of the intervention.
Response 7: Our study is a pioneer study that demonstrates that (i) the effects of spinal rTMS differ in dependence on coil positioning and (ii) the PP-rTSMS over the spinal cord is well tolerable and beneficial in supporting balance control. This approach should be investigated in disabled cohorts in the future. This fact is repeatedly mentioned in the manuscript.
Line 41: Conclusions: Our data show that PP-rTSMS effectively supports balance control and that coil orientation significantly influences these effects. Further studies should test variations of this promising approach on healthy and disabled cohorts.
Line 214: Despite the central role of the brain, the spinal cord is increasingly becoming the focus of noninvasive neuromodulation research. Several cohorts may benefit from this approach in the future. It has been repeatedly demonstrated that high-frequency rTSMS can induce neural, postural, and/or locomotor benefits in disabled cohorts [53-56]. For example, the application of both 10 Hz rTSMS [53] and 20 Hz rTSMS [54] over two weeks supported the recovery of neural tissue [53,54] together with locomotor recovery [53] after spinal cord injury. Ten sessions of 20 Hz rMS over C7 improved postural control and functional ambulation in patients with relapsing-remitting multiple sclerosis [55]. Eight sessions of 5 Hz rMS over Th12-L1 supported motor rehabilitation in Parkinson's disease [56]. Despite these encouraging results, the existing evidence is insufficient, and numerous additional studies are needed before noninvasive spinal modulation can be established in the framework of evidence-based medicine. Our study enhances the evidence in this field and shows that PP-rTSMS applied over L2 is safe, well-tolerated, and effective in supporting balance control in healthy people. Future studies should test this protocol in patient cohorts.
Line 268: The present study showed that PP-rTSMS is a safe, well-tolerated, and highly effective method for balance control. Future studies should investigate variations of this promising protocol in cohorts with different ages and health conditions. Patients with stroke, multiple sclerosis, spinal cord injury, Parkinson's disease, and other illnesses may benefit from this approach.
Comment 8:____________________________________________________________________
-The sample size is relatively small, which may limit the generalizability of the findings. You should describe with details how this sample size is determined.
Response 8: We included the determination of sample size in the Participants chapter. The limited generalizability of the finding is mentioned in the Conclusions chapter.
Line 135: Sample size calculation using G*power (version 3.1.9.7) analysis (effect size = 0.35, α error probability p < 0.05, Power = 0.95) indicated that a minimum of 27 subjects is required to detect statistically significant effects using ANOVA with four interventions and two time points.
Line 275: The generalizability of our findings is limited by relatively small sample size.
Comment 9:____________________________________________________________________
- Do the authors consider the level of physical activity of the participants in the inclusion criteria? This item is important factor in the neural network.
Response 9: Thank you for this input. Unfortunately, physical activity recording has not been performed in the framework of this experiment.
Comment 10:____________________________________________________________________
-Y balance test as a subjective test has not high accuracy for detection the balance and postural ability. The authors could use some postural tests such as the postural stability indices on Byodex system or force plate to assess the posture objectively.
Response 10: Thank you for this relevant hint. Indeed, it is believed that a simple Y-Balance-Test provides less accurate measures of balance control than technically sophisticated methods. However, the existing studies did not confirm this assumption. E.g., a systematic review demonstrates an excellent reliability of Y balance test. In contrast, the methodological quality of balance control measures on force plate was only insufficiently investigated up to now, and the existing data shows inconsistent results.
Powden CJ, Dodds TK, Gabriel EH. THE RELIABILITY OF THE STAR EXCURSION BALANCE TEST AND LOWER QUARTER Y-BALANCE TEST IN HEALTHY ADULTS: A SYSTEMATIC REVIEW. Int J Sports Phys Ther. 2019; 14: 683-694.
Troester JC, Jasmin JG, Duffield R. Reliability of Single-Leg Balance and Landing Tests in Rugby Union; Prospect of Using Postural Control to Monitor Fatigue. J Sports Sci Med. 2018; 17: 174-180.
Byrne A, Lodge C, Wallace J. Test-Retest Reliability of Single-Leg Time to Stabilization Following a Drop-Landing Task in Healthy Individuals. J Sport Rehabil. 2021; 30: 1242-1245.
Comment 11:____________________________________________________________________
Results
-The results are clearly presented, with appropriate use of statistical analysis to support the findings. Please add the mean age and rate of sex in the table 1 for each group and also report the between group analysis of these variables.
Response 11: This study used crossover study design and the probands description is thus identical for each group. The values are presented in the Results chapter.
Line 180: …. age 25.2 ± 3.4 years, 13 females, 14 males, 22 right-footed, 5 left-footed …
Comment 12:____________________________________________________________________
- it is better to consider a follow up assessing to investigate the long-term effects of interventions.
Response 12: Thank you for this hint. This information was included in the Conclusion chapter.
Line 278: Application of additional flow up evaluations may contribute to a better understanding of short and long term behavioural and neural effect.
Comment 13:____________________________________________________________________
-Please add the effect size of balance differences among group in the table 1.
Response 13: We performed statistical analysis for adverse effects in our study, in addition to balance task and neural processes evaluations. However, no significant effects were detected. Due to a big amount of the data, only significant p values and effects are presented in the manuscript. Thus, p values end effect size for adverse effects are missing.
Line 205: No severe adverse events were reported. The occurrence of fewer severe adverse events is shown in Table 2. No significant differences were detected between the conditions.
Comment 14:____________________________________________________________________
Discussion
The discussion needs to be interpreted the results and relates them to existing literature more details. Why the findings just indicated improvement of balance ability after intervention, without any changes in MEP and CSP. The reason of this findings should be explaining in the discussion. It seems that the interventions were not changes in neural network among groups. It may be to not shown any differences among groups even in the balance test, if this assessed long-termly.
Response 14: Thank you for this hint. We discuss the intervention-induced effects on balance task in detail on the pages 8-11. The absence of significant effects on MEP and CSP is also included.
Line 243: The MEP and CSP evaluations did not detect any effects on the corticospinal networks, which is inconsistent with previous observations [22-26]. Whether other neuroimaging techniques can be more successful in this regard remains an open question.
Comment 15:____________________________________________________________________
Please add the suggestions and limitations for future research. The potential impact of limitations of this study on the study's conclusions should be more explicitly discussed.
Response 15: This information is included in the conclusion chapter.
Line 44: Conclusions: Our data show that PP-rTSMS effectively supports balance control and that coil orientation significantly influences these effects. Further studies should test variations of this promising approach on healthy and disabled cohorts.
Line 275: The present study showed that PP-rTSMS is a safe, well-tolerated, and highly effective method for balance control. The generalizability of our findings is limited by the relatively small sample size. Future studies should investigate variations of this promising protocol in larger cohorts with different ages and health conditions. Patients with stroke, multiple sclerosis, spinal cord injury, Parkinson's disease, and other illnesses may benefit from this approach. Application of additional follow up evaluations may contribute to a better understanding of short and long term behavioural and neural effects.

Reviewer 2 Report
Comments and Suggestions for Authors
very interesting research:
did you register in clinicaltrial.org?
you have a pre-post experiment, mos appropiate analysis is ANCOVA adjusting by baseline, you have no repeated measures or better, an specific analysis for crosover designs:
Design and Analysis of Experiments with R. ByJohn Lawson. Edition1st Edition. First Published 2014. eBook Published17 December 2014. Pub. LocationNew York. ImprintChapman and Hall/CRC. DOIhttps://doi.org/10.1201/b17883, chapter 9
justify 5 days to wash up period
in table 2 add p values an effect size
asses normal distribution of your data to use parametrix or non-parametric
Author Response
Dear reviewer, thank you for the taking the time to read our manuscript and to make improvement suggestions. We revised the manuscript carefully in line with them. Our answers and modifications can be find below. We hope very much that our manuscript is now suitable for publication in Biomedicines.
Comment 1:_____________________________________________________________________
did you register in clinicaltrial.org?
Response 2: Thank you for this comment. The study was registered in the German Clinical Trail Register. We added this information in the Study design chapter.
Line 129: ... registered in the German Clinical Trial Register on 26 May 2025 (DRKS00037004).
Comment 2:_____________________________________________________________________
you have a pre-post experiment, mos appropiate analysis is ANCOVA adjusting by baseline, you have no repeated measures or better, an specific analysis for crosover designs: Design and Analysis of Experiments with R. ByJohn Lawson. Edition1st Edition. First Published 2014. eBook Published17 December 2014. Pub. LocationNew York. ImprintChapman and Hall/CRC. DOIhttps://doi.org/10.1201/b17883, chapter 9
Response 2: Thank your for this technical hint. We are the opinion that ANOVA is an appropriate measure to detect time*intervention interactions between four groups (grip cranially, grip caudally, grip laterally, sham) with two time points (pre, post).
Comment 3:_____________________________________________________________________
justify 5 days to wash up period
Response 3: Generally, a time frame of 24 hours is considered to be sufficient to avoid a carry over effects after a single rTMS application. The existing crossover studies use one, two or more days wash out period after a single session rTMS application. We decided to apply a few longer time frame to guarantee no appearance of carry over effects in our crossover study with four interventions.
Huang Y, Xia X, Meng X, Bai Y, Feng Z. Single session of intermittent theta burst stimulation alters brain activity of patients in vegetative state. Aging (Albany NY). 2024 Apr 18;16(8):7119-7130.
Veldema J, Nowak DA, Bösl K, Gharabaghi A. Hemispheric Differences of 1 Hz rTMS over Motor and Premotor Cortex in Modulation of Neural Processing and Hand Function. Brain Sci. 2023 May 2;13(5):752.
Parikh V, Medley A, Goh HT. Effects of rTMS to primary motor cortex and cerebellum on balance control in healthy adults. Eur J Neurosci. 2024 Jul;60(2):3984-3994.
Comment 4:_____________________________________________________________________
in table 2 add p values an effect size
Response 4: P values and effect size are included in the main text. Due to better clarity, only significant p-values and the corresponding effect sizes are presented.
Line 184: The total balance ability score improved significantly with real PP-rTSMS applied with all coil orientations (cranially (F1,26 = 8.009; p = 0.009; η² = 0.236), caudally (F1,26 = 4.846; p = 0.037; η² = 0.157), and laterally (F1,26 = 23,804; p ≤ 0.001; η² = 0.478) oriented grip) compared to sham stimulation. In addition, lateral coil grip orientation was associated with greater benefits than both cranial (F1,26 = 10.173; p = 0.004; η² = 0.281) and caudal (F1,26 = 14.058; p ≤ 0.001; η² = 0.351) coil grip orientations.
The balance ability of the right leg improved significantly with real PP-rTSMS applied with the coil handle oriented cranially (F1,26 = 4.821; p = 0.037; η² = 0.156) and laterally (F1,26 = 22.191; p ≤ 0.001; η² = 0.460) compared with sham stimulation. In addition, the lateral coil grip orientation was associated with a stronger improvement than the cranial (F1,26 = 7.344; p = 0.012; η² = 0.565) and caudal (F1,26 = 15.034; p ≤ 0.001; η² = 0.366) coil grip orientations.
The balance ability of the left leg improved significantly with real PP-rTSMS applied through a coil with a cranially (F1,26 = 9.486; p = 0.005; η² = 0.267), caudally (F1,26 = 7.382; p = 0.012; η² = 0.221), and laterally (F1,26 = 23.314; p ≤ 0.001; η² = 0.473) oriented grip compared to sham stimulation. In addition, the lateral coil grip orientation was associated with greater improvement than the cranial (F1,26 = 12,424; p = 0.002; η² = 0.323) and caudal (F1,26 = 8.966; p = 0.006; η² = 0.256) coil grip orientations.
Comment 5:_____________________________________________________________________
asses normal distribution of your data to use parametrix or non-parametric
Response 5: The normal distribution was assesses by Mauchly's sphericity tests. The Greenhouse-Geisser corrections were applied. This information is included in the Analysis chapter.
Line 176: Mauchly's sphericity tests and Greenhouse-Geisser corrections were applied.

Reviewer 3 Report
Comments and Suggestions for Authors
-------------------General Comments
Nevertheless, several clarifications are required, and I recommend that the authors proceed with caution when addressing these concerns. First, the statement presented in the penultimate paragraph of section 1.1 — “In the present study, we investigated the effects of spinal rMS in the same cohort” — raises significant concerns, which could be subject to criticism both from readers and the publisher. I am not implying that this constitutes salami science, but in order to avoid any perception of that, detailed clarifications are indeed necessary.
1) If this study was conducted on participants from the same cohort, it implies that the volunteers were similar to those enrolled in the previous study (Reference 16). As far as I could verify in the article published in Bioengineering, the inclusion criteria are nearly identical. This raises two issues. First, if participants were recruited from the same cohort, why are the inclusion criteria not exactly the same? For instance, the previous study stated “no relevant neurological, psychiatric, or orthopedic disorders”, whereas the current manuscript states “no relevant neurological, psychiatric, or orthopaedic disorders”. Why were neurological disorders not considered in this study?
2) Why was a sample size estimation performed in the previous study but not in the current one?
3) Why did the previous study include a greater number of functional tests, whereas the current study did not?
4) What is the rationale for not comparing DCS, ACS, and rMS in the present study?
5) Are the authors fully confident about the appropriateness of reusing the same image in two different publications (i.e., Figure 1 in the current manuscript appears identical to Figure 1, panel C, from the previous study)?
-------------------Specific Comments
-------------------Introduction
-Several vague statements are presented throughout the manuscript. For instance, the sentence “Other regions have rarely been considered” is unclear. Which regions are the authors referring to?
-------------------Methods
-This is also a major concern. The authors did not mention any guideline applicable to randomized studies, nor did they provide any information regarding trial registration — at least none that I could find.
-In addition to my disagreement with the use of p-value gradation to indicate levels of significance, the manuscript lacks confidence interval calculations and partial eta squared values.
-------------------Results
-The tables are extremely confusing. It is not possible to clearly identify what the values refer to, as all data are presented together without proper separation or labeling.
-------------------Discussion
-The results must be confined to the Results section. For example, Figure 2 appears in the Discussion section without any clear purpose and is not properly referenced or integrated into the text.
Author Response
Dear reviewer, thank you for the taking the time to read our manuscript and to make improvement suggestions. We revised the manuscript carefully in line with them. Our answers and modifications can be find below. We hope very much that our manuscript is now suitable for publication in Biomedicines.
Comment 1:_____________________________________________________________________
First, the statement presented in the penultimate paragraph of section 1.1 — “In the present study, we investigated the effects of spinal rMS in the same cohort” — raises significant concerns, which could be subject to criticism both from readers and the publisher. I am not implying that this constitutes salami science, but in order to avoid any perception of that, detailed clarifications are indeed necessary.
Response 1: Thank you for this hint. The present study did not include the same probands as our previous studies. We revised the manuscript to clear up this misunderstanding.
Line 83: In the present study, we investigated the effects of spinal rMS in a young healthy cohort.
Comment 2:_____________________________________________________________________
If this study was conducted on participants from the same cohort, it implies that the volunteers were similar to those enrolled in the previous study (Reference 16). As far as I could verify in the article published in Bioengineering, the inclusion criteria are nearly identical. This raises two issues. First, if participants were recruited from the same cohort, why are the inclusion criteria not exactly the same? For instance, the previous study stated “no relevant neurological, psychiatric, or orthopedic disorders”, whereas the current manuscript states “no relevant neurological, psychiatric, or orthopaedic disorders”. Why were neurological disorders not considered in this study?
Response 2: This study considered neurological disorders as exclusion criterion. This information is included in the Participants chapter.
Line 132: Individuals who met the following criteria were included: (1) age between 18 and 30 years, (2) no contraindications for noninvasive rMS [45]; and (3) no relevant neurological, psychiatric, or orthopaedic disorders.
Comment 3:_____________________________________________________________________
Why was a sample size estimation performed in the previous study but not in the current one?
Response 3: Thank you for this hint. We performed a sample size estimation in this study. This information was included in the Participants chapter.
Line 135: Sample size calculation using G*power (version 3.1.9.7) analysis (effect size = 0.35, α error probability p < 0.05, Power = 0.95) indicated that a minimum of 27 subjects is required to detect statistically significant effects using ANOVA with four interventions and two time points.
Comment 4:_____________________________________________________________________
Why did the previous study include a greater number of functional tests, whereas the current study did not?
Response 5: The study design, and the choice of assessments take in to account the fact, that the neural and behavioural rMS-induced changes dissipated gradually over the time. The existing studies indicates that strongest neural effects of a single session rMS are detectable up to 30 minutes after stimulation completing. Our current study included one behavioural assessment (Y Balance Test) and two electrophysiological assessments (MEP, CSP) that could be conducted within this time-frame.
Hoogendam, J.M.; Ramakers, G.M.; Di Lazzaro, V. Physiology of repetitive transcranial magnetic stimulation of the human brain. Brain Stimul. 2010, 3, 95–118.
Chung, S.W.; Hill, A.T.; Rogasch, N.C.; Hoy, K.E.; Fitzgerald, P.B. Use of theta-burst stimulation in changing excitability of motor cortex: A systematic review and meta-analysis. Neurosci. Biobehav. Rev. 2016, 63, 43–64.
Comment 5:_____________________________________________________________________
What is the rationale for not comparing DCS, ACS, and rMS in the present study?
Response 6: This study focused on rMS. A direct comparison with different stimulation techniques is surely also an attractive approach.
Comment 7:_____________________________________________________________________
Are the authors fully confident about the appropriateness of reusing the same image in two different publications (i.e., Figure 1 in the current manuscript appears identical to Figure 1, panel C, from the previous study)?
Response 7: In our option, both images shows several differences (see below, left is the prior publication’s image, the right from the current publication) and the figure is appropriate for publication.
Comment 8:_____________________________________________________________________
Introduction
-Several vague statements are presented throughout the manuscript. For instance, the sentence “Other regions have rarely been considered” is unclear. Which regions are the authors referring to?
Response 8: This is explained more closely in the following sentences.
Line 63: Although several neuroimaging studies have demonstrated that a wide range of other cortical and subcortical brain regions and the brainstem are crucially involved in balance control [8-9], existing technical solutions are insufficient to effectively target deep subcortical and brainstem structures. Therefore, the spinal cord is a promising alternative to the brain. Extensive data indicates that the spinal cord is crucially involved in balance tasks [10-12]. For example, the relationship between improved balance and suppressed Hoffmann reflex has been repeatedly described in both healthy and disabled cohorts [10-12]. A theory originating in the early twentieth century proposes that so called “central patterns generators” (CPGs) are core elements of posture and balance control [13-15]. These self-regulating neural circuits (located within the lumbar and cervical spinal cord intumescences) function under the control of the brainstem without input from higher brain regions. It is assumed that CPGs regulate optimal agonist–antagonist interactions during stereotypical rhythmic motor patterns (such as walking, cycling, or swimming) and dampen the disturbing effects of overshooting reflexive reactions (e.g. during stumbling) by facilitating or inhibiting alpha motoneuron activity [13-15].
Comment 9:_____________________________________________________________________
Methods
-This is also a major concern. The authors did not mention any guideline applicable to randomized studies, nor did they provide any information regarding trial registration — at least none that I could find.
Response 9: Thank you for this hint. We registered our trial in the German Clinical Trial Register and wrote the manuscript in line with recent CONSORT guidelines. We included this information in the Methods chapter.
Line 129: .... and registered in the German Clinical Trial Register on 26 May 2025 (DRKS00037004).
Line 130: The manuscript was written in accordance with the recent CONSORT guidelines [47].
Comment 10:_____________________________________________________________________
-In addition to my disagreement with the use of p-value gradation to indicate levels of significance, the manuscript lacks confidence interval calculations and partial eta squared values.
Response 10: We revised the interpretation and reporting of p value within the main text, tables and figures. Due to better clarity, we calculates effect sizes only for significant p-values and present them in the results chapter.
Line 173: P-values of ≤ 0.05 were considered statistically significant.
Line 494 / Table 1: S = significant intervention-induced changes compared to sham PP-rTSMS; * = P ≤ 0.05
Line 509 / Figure 2: * p ≤ 0.05
Line 37: Results: Each verum session induced a significant improvement in balance ability (cranially (F1,26 = 8.009; p = 0.009; η² = 0.236), caudally (F1,26 = 4.846; p = 0.037; η² = 0.157), and laterally (F1,26 = 23,804; p ≤ 0.001; η² = 0.478) oriented grip) as compared to the sham session. In addition, the laterally oriented coil grip was associated with significantly greater balance benefits than both the cranial (F1,26 = 10.173; p = 0.004; η² = 0.281) and caudal (F1,26 = 14.058; p ≤ 0.001; η² = 0.351) grip orientations.
Line 185: The total balance ability score improved significantly with real PP-rTSMS applied with all coil orientations (cranially (F1,26 = 8.009; p = 0.009; η² = 0.236), caudally (F1,26 = 4.846; p = 0.037; η² = 0.157), and laterally (F1,26 = 23,804; p ≤ 0.001; η² = 0.478) oriented grip) compared to sham stimulation. In addition, lateral coil grip orientation was associated with greater benefits than both cranial (F1,26 = 10.173; p = 0.004; η² = 0.281) and caudal (F1,26 = 14.058; p ≤ 0.001; η² = 0.351) coil grip orientations.
The balance ability of the right leg improved significantly with real PP-rTSMS applied with the coil handle oriented cranially (F1,26 = 4.821; p = 0.037; η² = 0.156) and laterally (F1,26 = 22.191; p ≤ 0.001; η² = 0.460) compared with sham stimulation. In addition, the lateral coil grip orientation was associated with a stronger improvement than the cranial (F1,26 = 7.344; p = 0.012; η² = 0.565) and caudal (F1,26 = 15.034; p ≤ 0.001; η² = 0.366) coil grip orientations.
The balance ability of the left leg improved significantly with real PP-rTSMS applied through a coil with a cranially (F1,26 = 9.486; p = 0.005; η² = 0.267), caudally (F1,26 = 7.382; p = 0.012; η² = 0.221), and laterally (F1,26 = 23.314; p ≤ 0.001; η² = 0.473) oriented grip compared to sham stimulation. In addition, the lateral coil grip orientation was associated with greater improvement than the cranial (F1,26 = 12,424; p = 0.002; η² = 0.323) and caudal (F1,26 = 8.966; p = 0.006; η² = 0.256) coil grip orientations.
Comment 11:_____________________________________________________________________
Results
-The tables are extremely confusing. It is not poble to clearly identify what the values refer to, as all data are presented together without proper separation or labeling.
Response 11: Thank you for this hint. We added line grids to improve the clarity of our tables.
Comment 12:_____________________________________________________________________
Discussion
-The results must be confined to the Results section. For example, Figure 2 appears in the Discussion section without any clear purpose and is not properly referenced or integrated into the text.
Response 12: The Table 2 is included in the Results section (3.3. Adverse effects)
Line 199: The occurrence of fewer severe adverse events is shown in Table 2.

Round 2
Reviewer 2 Report
Comments and Suggestions for Authors
paper is now ready, only please explain if you used ks or shapiro-wilk test to asses normality of data
Author Response
Comment:_______________________________________________________________________
Paper is now ready, only please explain if you used ks or shapiro-wilk test to asses normality of data.
Answer: Thank you for your positive evaluation of our manuscript. The information requested was included in the Analysis chapter. Ther change is marked in red and yellow.
The Kolmogorov-Smirnov normality test, Mauchly's sphericity tests and Greenhouse-Geisser corrections were applied.

Reviewer 3 Report
Comments and Suggestions for Authors
I consider the responses insufficient and maintain my comments.
Author Response
Dear reviewer we have revised the first version of our manuscript very carefully and have spent a lot of time with it. So, we are unpleasantly surprised by your short and dismissive reply “I consider the responses insufficient and maintain my comments”. We wonder whether is this a misunderstanding? Your comments regarding the introduction (comment 8), cohort description (comment 1,2), sample size estimation (comment 3), guidelines using (comment 9), p-value interpretation (comment 9), assessments (comment 4), stimulation technique (comment 5), figures (comment 6) and tables (comment 10) were responded to and implemented in the framework of this manuscript. Your comments, our responses and the manuscript´s modifications are listed below.
Comment 1:_____________________________________________________________________
First, the statement presented in the penultimate paragraph of section 1.1 — “In the present study, we investigated the effects of spinal rMS in the same cohort” — raises significant concerns, which could be subject to criticism both from readers and the publisher. I am not implying that this constitutes salami science, but in order to avoid any perception of that, detailed clarifications are indeed necessary.
Response 1: Thank you for this hint. The present study did not include the same probands as our previous studies. We revised the manuscript to clear up this misunderstanding.
Line 83: In the present study, we investigated the effects of spinal rMS in a young healthy cohort.
Comment 2:_____________________________________________________________________
If this study was conducted on participants from the same cohort, it implies that the volunteers were similar to those enrolled in the previous study (Reference 16). As far as I could verify in the article published in Bioengineering, the inclusion criteria are nearly identical. This raises two issues. First, if participants were recruited from the same cohort, why are the inclusion criteria not exactly the same? For instance, the previous study stated “no relevant neurological, psychiatric, or orthopedic disorders”, whereas the current manuscript states “no relevant neurological, psychiatric, or orthopaedic disorders”. Why were neurological disorders not considered in this study?
Response 2: This study considered neurological disorders as exclusion criterion. This information is included in the Participants chapter.
Line 132: Individuals who met the following criteria were included: (1) age between 18 and 30 years, (2) no contraindications for noninvasive rMS [45]; and (3) no relevant neurological, psychiatric, or orthopaedic disorders.
Comment 3:_____________________________________________________________________
Why was a sample size estimation performed in the previous study but not in the current one?
Response 3: Thank you for this hint. We performed a sample size estimation in this study. This information was included in the Participants chapter.
Line 135: Sample size calculation using G*power (version 3.1.9.7) analysis (effect size = 0.35, α error probability p < 0.05, Power = 0.95) indicated that a minimum of 27 subjects is required to detect statistically significant effects using ANOVA with four interventions and two time points.
Comment 4:_____________________________________________________________________
Why did the previous study include a greater number of functional tests, whereas the current study did not?
Response 5: The study design, and the choice of assessments take in to account the fact, that the neural and behavioural rMS-induced changes dissipated gradually over the time. The existing studies indicates that strongest neural effects of a single session rMS are detectable up to 30 minutes after stimulation completing. Our current study included one behavioural assessment (Y Balance Test) and two electrophysiological assessments (MEP, CSP) that could be conducted within this time-frame.
Hoogendam, J.M.; Ramakers, G.M.; Di Lazzaro, V. Physiology of repetitive transcranial magnetic stimulation of the human brain. Brain Stimul. 2010, 3, 95–118.
Chung, S.W.; Hill, A.T.; Rogasch, N.C.; Hoy, K.E.; Fitzgerald, P.B. Use of theta-burst stimulation in changing excitability of motor cortex: A systematic review and meta-analysis. Neurosci. Biobehav. Rev. 2016, 63, 43–64.
Comment 5:_____________________________________________________________________
What is the rationale for not comparing DCS, ACS, and rMS in the present study?
Response 6: This study focused on rMS. A direct comparison with different stimulation techniques is surely also an attractive approach.
Comment 7:_____________________________________________________________________
Are the authors fully confident about the appropriateness of reusing the same image in two different publications (i.e., Figure 1 in the current manuscript appears identical to Figure 1, panel C, from the previous study)?
Response 7: In our option, both images shows several differences (see below, left is the prior publication’s image, the right from the current publication) and the figure is appropriate for publication.
Comment 8:_____________________________________________________________________
Introduction
-Several vague statements are presented throughout the manuscript. For instance, the sentence “Other regions have rarely been considered” is unclear. Which regions are the authors referring to?
Response 8: This is explained more closely in the following sentences.
Line 63: Although several neuroimaging studies have demonstrated that a wide range of other cortical and subcortical brain regions and the brainstem are crucially involved in balance control [8-9], existing technical solutions are insufficient to effectively target deep subcortical and brainstem structures. Therefore, the spinal cord is a promising alternative to the brain. Extensive data indicates that the spinal cord is crucially involved in balance tasks [10-12]. For example, the relationship between improved balance and suppressed Hoffmann reflex has been repeatedly described in both healthy and disabled cohorts [10-12]. A theory originating in the early twentieth century proposes that so called “central patterns generators” (CPGs) are core elements of posture and balance control [13-15]. These self-regulating neural circuits (located within the lumbar and cervical spinal cord intumescences) function under the control of the brainstem without input from higher brain regions. It is assumed that CPGs regulate optimal agonist–antagonist interactions during stereotypical rhythmic motor patterns (such as walking, cycling, or swimming) and dampen the disturbing effects of overshooting reflexive reactions (e.g. during stumbling) by facilitating or inhibiting alpha motoneuron activity [13-15].
Comment 9:_____________________________________________________________________
Methods
-This is also a major concern. The authors did not mention any guideline applicable to randomized studies, nor did they provide any information regarding trial registration — at least none that I could find.
Response 9: Thank you for this hint. We registered our trial in the German Clinical Trial Register and wrote the manuscript in line with recent CONSORT guidelines. We included this information in the Methods chapter.
Line 129: .... and registered in the German Clinical Trial Register on 26 May 2025 (DRKS00037004).
Line 130: The manuscript was written in accordance with the recent CONSORT guidelines [47].
Comment 10:_____________________________________________________________________
-In addition to my disagreement with the use of p-value gradation to indicate levels of significance, the manuscript lacks confidence interval calculations and partial eta squared values.
Response 10: We revised the interpretation and reporting of p value within the main text, tables and figures. Due to better clarity, we calculates effect sizes only for significant p-values and present them in the results chapter.
Line 173: P-values of ≤ 0.05 were considered statistically significant.
Line 494 / Table 1: S = significant intervention-induced changes compared to sham PP-rTSMS; * = P ≤ 0.05
Line 509 / Figure 2: * p ≤ 0.05
Line 37: Results: Each verum session induced a significant improvement in balance ability (cranially (F1,26 = 8.009; p = 0.009; η² = 0.236), caudally (F1,26 = 4.846; p = 0.037; η² = 0.157), and laterally (F1,26 = 23,804; p ≤ 0.001; η² = 0.478) oriented grip) as compared to the sham session. In addition, the laterally oriented coil grip was associated with significantly greater balance benefits than both the cranial (F1,26 = 10.173; p = 0.004; η² = 0.281) and caudal (F1,26 = 14.058; p ≤ 0.001; η² = 0.351) grip orientations.
Line 185: The total balance ability score improved significantly with real PP-rTSMS applied with all coil orientations (cranially (F1,26 = 8.009; p = 0.009; η² = 0.236), caudally (F1,26 = 4.846; p = 0.037; η² = 0.157), and laterally (F1,26 = 23,804; p ≤ 0.001; η² = 0.478) oriented grip) compared to sham stimulation. In addition, lateral coil grip orientation was associated with greater benefits than both cranial (F1,26 = 10.173; p = 0.004; η² = 0.281) and caudal (F1,26 = 14.058; p ≤ 0.001; η² = 0.351) coil grip orientations.
The balance ability of the right leg improved significantly with real PP-rTSMS applied with the coil handle oriented cranially (F1,26 = 4.821; p = 0.037; η² = 0.156) and laterally (F1,26 = 22.191; p ≤ 0.001; η² = 0.460) compared with sham stimulation. In addition, the lateral coil grip orientation was associated with a stronger improvement than the cranial (F1,26 = 7.344; p = 0.012; η² = 0.565) and caudal (F1,26 = 15.034; p ≤ 0.001; η² = 0.366) coil grip orientations.
The balance ability of the left leg improved significantly with real PP-rTSMS applied through a coil with a cranially (F1,26 = 9.486; p = 0.005; η² = 0.267), caudally (F1,26 = 7.382; p = 0.012; η² = 0.221), and laterally (F1,26 = 23.314; p ≤ 0.001; η² = 0.473) oriented grip compared to sham stimulation. In addition, the lateral coil grip orientation was associated with greater improvement than the cranial (F1,26 = 12,424; p = 0.002; η² = 0.323) and caudal (F1,26 = 8.966; p = 0.006; η² = 0.256) coil grip orientations.
Comment 11:_____________________________________________________________________
Results
-The tables are extremely confusing. It is not poble to clearly identify what the values refer to, as all data are presented together without proper separation or labeling.
Response 11: Thank you for this hint. We added line grids to improve the clarity of our tables.
Comment 12:_____________________________________________________________________
Discussion
-The results must be confined to the Results section. For example, Figure 2 appears in the Discussion section without any clear purpose and is not properly referenced or integrated into the text.
Response 12: The Table 2 is included in the Results section (3.3. Adverse effects)
Line 199: The occurrence of fewer severe adverse events is shown in Table 2.
